# Executive functions form a single construct and are associated with schooling: Evidence from three low- and middle- income countries

**Charlotte Wray**[1], **Alysse Kowalski**[2], **Feziwe Mpondo**[3], **Laura Ochaeta**[4], **Delia Belleza**[5], **Ann DiGirolamo**[6], **Rachel Waford**[7], **Linda Richter**[3], **Nanette Lee**[8], **Gaia Scerif**[9], **Aryeh D. Stein**[7], **Alan Stein**[1]*, **COHORTS**[¶]

1 Department of Psychiatry, The University of Oxford, Oxford, United Kingdom, 2 Nutrition and Health Sciences Program, Laney Graduate School, Emory University, Atlanta, GA, United States of America, 3 DSI-NRF Centre of Excellence in Human Development at the University of the Witwatersrand, Johannesburg, South Africa, 4 Institute of Nutrition of Central America and Panama, Guatemala City, Guatemala, 5 Department of Psychology, School of Arts and Sciences, University of San Carlos, Cebu, Philippines, 6 Georgia Health Policy Center, Georgia State University, Atlanta, GA, United States of America, 7 Hubert Department of Global Health, Rollins School of Public Health, Emory University, Atlanta, GA, United States of America, 8 USC-Office of Population Studies Foundation, Inc., University of San Carlos, Cebu, The Philippines, 9 Department of Experimental Psychology, The University of Oxford, Oxford, United Kingdom

¶ Membership of the COHORTS Group is provided in the Acknowledgments.
* Alan.Stein@psych.ox.ac.uk

**Data Availability Statement:** The data used in this study are from active cohort studies with previously published recruitment information and hence there is potential for individual participant re-

## Abstract

Measuring executive function (EF) among adults is important, as the cognitive processes involved in EF are critical to academic achievement, job success and mental health. Current evidence on measurement and structure of EF largely come from Western, Educated, Industrialized, Rich and Democratic (WEIRD) countries. However, measuring EF in low-and-middle-income countries (LMICs) is challenging, because of the dearth of EF measures validated across LMICs, particularly measures that do not require extensive training, expensive equipment, or professional administration. This paper uses data from three LMIC cohorts to test the feasibility, validity and reliability of EF assessment in adults using three sub-tests (representing key components of EF) of the NIH Toolbox Cognitive battery. For each cohort, all three EF measures (inhibition, flexibility and working memory) loaded well onto a unidimensional latent factor of EF. Factor scores related well to measures of fluid intelligence, processing speed and schooling. All measures showed good test-retest reliability across countries. This study provides evidence for a set of sound measures of EF that could be used across different cultural, language and socio-economic backgrounds in future LMIC research. Furthermore, our findings extend conclusions on the structure of EF beyond those drawn from WEIRD countries.

## Introduction

Executive Functions (EF) are cognitive control processes which facilitate decision making, planning and goal-driven behaviour. EFs include inhibitory control, working memory and

identification if the raw data are made freely available. Data are available to qualified researchers upon request, subject to confidentiality agreements. Data requests can be made to the following: South Africa data: Linda Richter (Linda. Richter@wits.ac.za) or Shane Norris (Shane. Norris@wits.ac.za). Guatemala data: Manuel Ramirez (mramirez@incap.int) or Dina Roche (droche@incap.int). Philippines data: USC Office of Population Studies Foundation (opsfoundation@opsusc.org), Nanette Lee (nanette_rlee@yahoo.com) or Delia Carba (carbadel@yahoo.com).

**Funding:** This study was funded by the Bill and Melinda Gates Foundation (OPP1164115). The funders had no role in study design, data collection and analysis, decision to publish, or preparation of the manuscript.

**Competing interests:** The authors have declared that no competing interests exist.

cognitive flexibility [1]; which are crucial to academic attainment, cognitive development, mental well-being [2], and optimising human capital potential. For example, EF abilities predict school readiness [3, 4], achievement on academic tests [5, 6], as well as job success [7] and quality of life [8, 9].

Early life experiences such as disease, malnutrition and violence can adversely impact brain development [10], especially affecting cognitive skills such as executive functions. This is particularly important in low- and middle-income countries (LMIC) where the prevalence of adverse early life experiences, compromising cognition, in some contexts is high [11].

Miyake and colleagues indicate both unity and diversity within EF models, whereby inhibition, cognitive flexibility and working memory are correlated with each other and form a unitary EF factor but are also separable (diversity) [1, 12]. However, such cognitive models of EFs have been primarily based on research from Western, Educated, Industrialized, Rich and Democratic (WEIRD) countries [13]. Fundamentally, this is a limitation to the generalisability of cognitive models more broadly, and those of EF in particular, because it remains unclear whether they apply beyond WEIRD contexts.

In addition to understanding how EF skills can be measured and relate to one another in LMICs, it is also important to consider factors which are associated with EF in LMICs; such factors, if modifiable, could be critical in developing interventions. One such factor is schooling, which studies have shown to be associated with EF in childhood and adolescence [3–6, 14–16]. Again, less is known about these associations in LMICS, nor associations between adult EF and schooling. Having robust measures of EF, and identifying associated factors such as schooling, is critical to fully understand the pathways to optimal outcomes in adulthood and thus inform intervention.

Furthermore, identifying reliable methods to assess EF that can be administered by non-specialist staff is critical, to ensure reach and representation of communities without access to specialists. Measuring EF in LMICs is fundamentally challenging, as the majority of existing assessments were developed and validated in high income countries (HICs) and many require in-depth training [17]. In addition, studies which use EF measurement tools have predominantly been conducted in HICs and often do not report reliability and/or validity [18]. As a consequence, there is limited evidence of the robustness and reliability of EF measures, particularly in LMICs.

Furthermore, using EF measures developed in HICs may lead to interpretation issues when used in LMIC contexts. Semrud-Clikeman et al. [17] highlight that many cognitive tests do not necessarily measure the same underlying constructs when administered in different cultures, as tasks which are familiar to people in HICs may not be familiar to those in LMICs. For example, for some participants, working with numbers or puzzles may be novel, especially for those with limited educational experience. As a result, normative data for measures developed in HICs often do not generalise to LMICs as they tend to over-estimate cognitive delay (See Semrud-Clikeman et al., 2017). To counteract this, adaptations are often made to tasks to minimise cultural bias. The extent to which measures developed in HICs are appropriate for measuring EF in different countries and cultures, even with adaptations, requires investigation.

The aim of the current study was to investigate the extent to which three measures of EF, developed in HICs, are reliable indicators of EF in LMICs and the extent to which they are associated with key indicators of human capital, such as schooling.

We utilised the NIH Toolbox because their EF-focused assessments correspond to the three core EFs: inhibition, cognitive flexibility and working memory [1]. In addition, these measures have been well validated, showing good convergent and discriminant validity and good test-retest reliability among both adults and children in the US [19, 20]. The NIH Toolbox development team considered cultural and linguistic issues during its development as well as cultural

diversity, albeit within the US population [21]. Thus, less is known about how these measures generalise across countries, cultures and languages other than the USA. We assessed the cultural relevance of the NIH Toolbox tests and adapted them where necessary to ensure they were culturally appropriate.

We used a round of standardised data collection in adulthood from three LMIC birth cohorts (Guatemala, Philippines, and South Africa), as an opportunity to test the suitability, validity and reliability of tablet-based, field worker friendly assessments across diverse LMICs. These cohorts vary in language, culture, climate and diversity. Each birth cohort is part of the Consortium of Health-Orientated Research in Transitioning Societies (COHORTS). These large cohort studies have prospectively followed participants throughout their lives and this paper utilises data collected when they were re-visited in adulthood. Participants have grown up in countries which have experienced rapid social, economic and environmental changes and some have grown up in environments which restrict human potential.

These three diverse LMIC birth cohorts offer a unique opportunity to explore the feasibility, validity and reliability of EF assessments through exploring the factor structure and test-retest reliability of these measures. Crucially, they enable us to examine executive function indicators as specific correlates of key human capital variables, such as schooling. In addition, they enable us to explore their relationship to other indices of cognition, such as fluid intelligence [22, 23] and processing speed [24, 25], which are known to correlate with EF in HICs. Such research is essential before EF measurement tools can be reliably used in LMICs. Similar factor structures, relations to other indicators and good test-retest reliability in all three countries would provide sound evidence for the effectiveness of these measures across diverse countries and cultures, supporting the use of the NIH Toolbox to assess adult EF in cross-cultural research. Given that many people in LMICs are much more likely to be exposed to disadvantage and other risk factors, which can compromise EF, identifying robust measures of EF and associated factors is critical.

## Method

### Participants

**Institute of nutrition of central America and Panama nutrition trial cohort (INCAP), Guatemala.** The INCAP study began in 1969 and originally comprised 2392 children born between from 1962–1977 in eastern Guatemala (See Stein et al. [26] for cohort profile). This paper reports data collected when cohort participants were 40–57 years old (n = 1271).

For the reliability study forty-five additional participants, who were not part of the INCAP cohort, were recruited from a community that demographically resembled the main cohort communities (38–58 years old).

**Cebu Longitudinal Health and Nutrition Survey (CLHNS), Philippines.** The CLHNS recruited a community-based sample of pregnant women in 1983–1984 in Metropolitan Cebu, Philippines. Participants were followed up throughout infancy, childhood and adulthood (N = 3080 at recruitment); see Adair et al. [27] for cohort profile. This paper reports data collected when cohort participants were 34–36 years old (n = 1327).

Thirty-two additional participants, who were not part of the CLHNS cohort, were recruited for the reliability study. Inclusion criteria for the reliability assessment required participants to have a similar background to that of the CLHNS cohort, in terms of age and socio demographic profile (aged 32–36 years).

**Birth to Twenty Plus (Bt20+), Soweto, South Africa.** Established in 1990, the Bt20 + cohort comprised 3273 singleton births enrolled within a 6-week period in public hospitals in Soweto, Johannesburg. The study sample has been prospectively followed from birth to

adulthood (See Richter et al. [28] for cohort profile). This paper reports data collected when cohort participants were 28–29 years old (n = 1402).

Forty-three participants from the main BT20+ cohort, also participated in the reliability study. All 43 participants completed the first and second assessments and 30 completed the third assessment.

## Adaptations and translation

Before the study began, the cultural appropriateness of all tasks was assessed. As a result, some tests required adaptation, which are discussed below.

**Developing the Cebuano NIH Toolbox app.** This study used the English and Spanish versions of the NIH Toolbox app in South Africa and Guatemala, respectively. As these versions could not be used in the Philippines, in conjunction with the NIH toolbox team, a Cebuano version of the app for administration in Philippines was developed. This process involved forward and back translation of all instructions and items. A linguist and a licensed psychologist compared the original item, the Cebuano version and the back translation for consistency. Any discrepancies were reviewed and revised as necessary. Finally, a language co-ordinator checked the proposed Cebuano version, for acceptability and appropriateness for the target population.

Once the translations were approved, audio-recordings were created. The NIH Toolbox team then processed the audio recordings and developed the Cebuano version of the app [29]. Finally, the research team at the USC-Office of Population Studies Foundation checked the new Cebuano app to ensure that all stimuli and instructions matched the audio recorded script and the timing of audio and visual instruction and/or pictures was correct.

**Adaptations.** Item substitutions were made to the list sorting stimuli in Guatemala and Philippines. Items which were not common within these countries were substituted for more common items, to ensure participants knew the target items. As a key component of the list sorting task is the size of each item, replacement items were the same size as the original item. In Guatemala, *pumpkin* was substituted with *papaya*, *cherry* with *nispero* (loquat) and *blueberry* with *nance* (small round tropical fruit). In Philippines, *pumpkin* was substituted with *kalabasa* (squash), *blueberry* with *lomboy* (java plum), *cherry* with *sereguelas* (Spanish plum) and *peach* with *santol* (cotton fruit). All adaptations and translations were approved and implemented by the NIH Toolbox development team.

## Procedure

**Main study.** Participants were assessed by a trained field worker over a two-hour period, the survey instruments included cognitive assessments, and interviewer administered questionnaires. The cognitive assessments comprised three EF-focused assessments from the NIH Toolbox cognitive battery, administered using an iPad. In addition, measures of processing speed and non-verbal IQ were also administered as fluid intelligence and processing speed are closely related to EF. All cognitive assessments were conducted at the start of the session, to prevent fatigue. The tasks were administered in Spanish in Guatemala, in Cebuano in the Philippines and English in South Africa.

All field workers were recruited locally and completed training in administration of cognitive assessments before the start of the study and refresher training halfway through data collection. The educational backgrounds of the field teams range from secondary school education to master's degrees in psychology. Regular observations were carried out to ensure fidelity and quality control across sites.

Written consent was obtained from all participants before the assessment began. Ethics clearance for the study was obtained from the institutional review boards of Emory University,

Atlanta, USA; INCAP, Guatemala City, Guatemala; University of San Carlos, Cebu, Philippines; University of the Witwatersrand, Johannesburg, South Africa and the University of Oxford.

**Test retest study.** Participants completed the same assessments as the participants in the main study, however the test-retest study participants completed the assessments on multiple occasions. Participants were seen by the same field worker at each time point. In Guatemala and Philippines, the tasks were administered twice, on two different days (Guatemala mean interval: 21 days, range: 17–23 days; Philippines mean interval: 8 days, range: 7–17 days). In South Africa, the tasks were administered three times, on three different days (Mean interval T1-T2: 57 days, range: 2–191 days; Mean Interval T2-T3: 14 days, range: 13–21 days).

## Measures

**Executive function measures.** *NIH Toolbox Flanker inhibitory control and attention [30].* The Flanker task measures inhibition and attentional control by asking participants to attend to target stimuli, whilst inhibiting information irrelevant to the task goals. During each trial, participants are presented with a row of five arrows and asked to indicate the direction of the <u>middle</u> arrow, by selecting an icon pointing in the correct direction. Participants complete 20 trials in total; twelve congruent trials (the flanking arrows all point in the same direction as the middle arrow) and eight incongruent trials (the flanking arrows point in the opposite direction to the middle arrow). Accuracy and reaction time are both recorded. The NIH Toolbox uses a two-vector algorithm which integrates accuracy and reaction time scores, to create a computed score.

*NIH Toolbox Dimensional Change Card Sort (DCCS) [30].* The DCCS task measures cognitive flexibility and attention by asking participants to switch between matching pictures by colour and matching pictures by shape. Participants are given a cue before each trial, to indicate if they should match by colour or shape. Participants complete 30 test trials, which comprise 23 repeat trials and 7 switch trials. Accuracy and reaction time are recorded for all trials. The same two-vector algorithm, which integrates accuracy and reaction time scores, is used for the DCCS task to create a computed score.

*NIH Toolbox List Sorting working memory [30].* Working memory was measured through the NIH Toolbox List Sorting task. During this task, participants are presented aurally and visually with lists of either foods, animal or both. They are asked to recall lists, in size order from smallest to largest. The number of correctly recalled lists is summed to create a total score, the maximum score for this task is 26. Participants respond orally, and answers are recorded as correct or incorrect by the examiner.

**Additional measures.** *Processing speed: NIH Toolbox pattern comparison processing speed test [30].* The Pattern Comparison Task measures processing speed by asking participants to make quick judgements about whether two stimuli are the same. If the stimuli are the same, the participant responds by pressing the "yes" icon; if the pictures are not the same, they press the "no" icon. The total score comprises the number of correct trials participants are able to complete within 90 seconds (maximum 130).

*Fluid intelligence: Raven's Standard Progressive Matrices [31].* Participants also completed the Raven's Standard Progressive Matrices, a measure of fluid intelligence. Participants are presented with a pattern, for which there is a piece missing. Participants are asked to select which piece completes the pattern from a number of options below the pattern. The testing booklet comprises five sections (A-E), which increase in difficulty. In Philippines and South Africa, participants were asked to complete as many problems as they could in 30 minutes (Maximum score 60). In Guatemala, only section A-C were administered (maximum score

36), as previous studies with the Guatemala cohort indicated participants rarely progressed past section C.

*Schooling*: *Highest grade attained*. In each cohort, self-reported information regarding schooling was collected, which was defined here as the highest grade attained.

## Analysis

Data completeness are reported as an indicator of the feasibility of participants completing all of the assessment. Large numbers of missing data may indicate barriers to completing the assessments, which may impact the extent to which these measures can reliably measure EF in LMICs.

Given the uncertainty in the degree to which the unitary model of EF represented by the literature in HICs [1, 12] would be applicable in LMICs, we first pooled data across the sites and conducted an exploratory factor analysis using scores on the Flanker (Inhibition), DCCS (Cognitive Flexibility) and List Sorting (Working Memory) tasks, which index executive function. We used parallel analysis to inform the number of factors in the EFA. Using the factor structure identified in the EFA, we conducted a confirmatory factor analysis and used full information maximum likelihood to estimate model parameters while accounting for missing data. We considered chi-square $>0.05$, root mean squared residual (RMSEA) $<0.05$, comparative fit index (CFI) $>0.95$, Tucker Lewis Index (TLI) $>0.95$, and standardized root mean square residual (SRMR) $<0.08$ indicators of good model fit [32]. The CFA model was assessed for weak invariance by site using a chi-square test and examining change in CFI ($<0.01$) and RMSEA ($<0.015$). The factor loadings were not invariant by site; therefore, we repeated the EFA and CFA procedures by site to identify site-specific latent EF structures. Factor scores from the site-specific EF models were generated.

We explored the relationship between an EF latent factor score, and non-verbal IQ, processing speed and schooling, through Pearson correlation analysis. To further investigate the reliability of the NIH Toolbox, practice effects were evaluated using t-tests and effect sizes. Cohens d effect sizes are reported as an indication of the magnitude of practice effects and interpreted as 0.2 being a small effect, 0.5 a moderate effect and 0.8 a large effect [33]. Finally, test-retest reliability was evaluated using Intraclass correlation. Intra-class correlation coefficients interpreted as: $<0.5$ poor, 0.50–0.75 moderate, 0.75–0.90 good and above 0.90 excellent test-retest reliability [34].

## Results

### Main study

Across the three sites, EF and additional cognitive data were available for over 98% of participants, with over 95% of these participants completing all five assessments. The primary reasons for failure to complete individual tests were technical issues, and failure to pass beyond the practice trials. Means and SD of the scores of completed tests are provided in Table 1.

**Correlations between executive function measures.** As demonstrated by Table 2 measures of Inhibition (Flanker), cognitive flexibility (DCCS) and working memory (List Sort) were all significantly correlated with each other in each site.

**Factor analysis: A single EF factor.** The parallel analysis of inhibition, flexibility, and working memory scores suggested a single factor best fit the data. We specified an EFA with one latent factor and advanced this solution to a CFA. A single latent factor with three indicators is just-identified and therefore has perfect fit, limiting the utility of traditional fit indices and modification indices to improve model fit. Weak invariance by site could not be established (see Table 3). Following this we then repeated the EFA and CFA analyses by site, developing a latent factor structure of EF for each site. The results of the EFA and CFA within each

**Table 1. Means scores (SD) for each measure, in each cohort.**

| Assessment | | Guatemala (N = 1247) | | Philippines (n = 1327) | | South Africa (n = 1327) | |
|---|---|---|---|---|---|---|---|
| | | N | Mean (SD) | N | Mean (SD) | N | Mean (SD) |
| Executive Function | Inhibition (Flanker Computed Score) | 1213 | 5.56 (1.19) | 1327 | 7.48 (1.08) | 1327 | 7.43 (1.11) |
| | Cognitive Flexibility (DCCS Computed score) | 1223 | 5.26 (1.97) | 1327 | 7.53 (1.29) | 1327 | 7.52 (1.18) |
| | Working Memory (List Sort raw score) | 1200 | 12.06 (3.78) | 1284 | 14.43 (3.68) | 1306 | 14.72 (3.20) |
| Additional Measures | Processing Speed (Pattern Comparison raw score) | 1240 | 31.59 (8.56) | 1326 | 39.43 (7.56) | 1327 | 41.80 (7.78) |
| | Fluid Intelligence (RSPM raw score) | 1212 | 16.40 (5.59) | 1327 | 32.15 (11.30) | 1310 | 37.00 (9.82) |
| | Schooling (highest grade) | 1244 | 5 (4) | 1325 | 11 (3) | 1303 | 12 (1) |

Note: Guatemala only administered Raven's A-C (Max score = 36), Philippines and South Africa administered A-E (Max score = 60). RSPM: Raven's Standard Progressive Matrices; DCCS: Dimensional Change Card Sorting.

site were consistent with the pooled analysis; covariances between the inhibition, flexibility, and working memory measures were best described by a single latent of EF. In site-specific models in Guatemala the one factor model had an eigen value of 1.50 and accounted for 50% of the variance in factor score. In the Philippines the one factor model had an eigen value of 1.37 and accounted for 46% of the variance in factor score. In South Africa the one factor model had an eigen value of 1.26 and accounted for 42% of the variance in factor score. See Fig 1 for the factor loadings in each country.

**Correlating EF with other cognitive and schooling measures.** We next explored how the EF factor score related to the other cognitive measures and schooling. The EF factor score was significantly correlated with speed of processing, non-verbal IQ and schooling, for all three sites (see Table 4).

## Test- retest reliability study

The following analyses used participants who were not part of the main cohort in Guatemala and Philippines, and who were not included in the above factor analyses. In South Africa, Bt20 + cohort participants were used for the test-retest study and their T1 data was included in the above analysis.

**Table 2. Correlations between each measure across each site.**

| Country | Measure | Flanker | DCCS |
|---|---|---|---|
| Guatemala | DCCS | 0.54** | 1 |
| | List Sort | 0.41** | 0.48** |
| Philippines | DCCS | 0.57** | 1 |
| | List Sort | 0.32** | 0.36** |
| South Africa | DCCS | 0.54** | 1 |
| | List Sort | 0.29** | 0.32** |

**p<0.001.

**Table 3. Measurement invariance by site.**

| Model | $\chi^2$ | df | $p > \chi^2$ | CFI | ΔCFI | RMSEA | ΔRMSEA | BIC |
|---|---|---|---|---|---|---|---|---|
| Pooled Sample | 0.00 | 0 | - - | 1.00 | -- | 0.00 | -- | 46590 |
| 1. configural | 0.00 | 0 | -- | 1.00 | -- | 0.00 | -- | 44055 |
| 2. loadings | 34.42 | 4 | <0.01 | 0.99 | 0.014 | 0.077 | 0.077 | 44056 |

**Practice effects.** *Guatemala*. Participants from the Guatemala sample exhibited better performance during the retest session than the first test session across all tasks (See Table 5 for means). There were small significant practice effects between T1 and T2 for all tasks (see Table 5).

*Philippines*. Within the Philippines sample, participants performed better during the retest than during the initial test (see Table 6 for means). There were small significant practice effects between T1 and T2 for the DCCS and Raven's Progressive Matrices. There were small but non-significant practice effects between T1 and T2 for The Flanker computed score, List Sort and Pattern Comparison.

*South Africa*. There were small significant practice effects between T1 and T2 for The Flanker computed score, and List Sort. In addition, small non-significant practice effects were observed between T1 and T2 for the DCCS, Pattern comparison, and Raven's Standard Progressive Matrices (see Table 7 for means).

There were small significant practice effects between T2 and T3 for the Pattern comparison and small non-significant practice effects between T2 and T3 for the Flanker, DCCS, List sort and Raven's Standard Progressive Matrices (see Table 7 for means).

**Test-retest reliability.** Next, we considered test-retest reliability of each of the measures through Intraclass correlation analysis. As demonstrated by Table 8, a high degree of reliability was observed for all measures across all three countries. However, there were a couple of

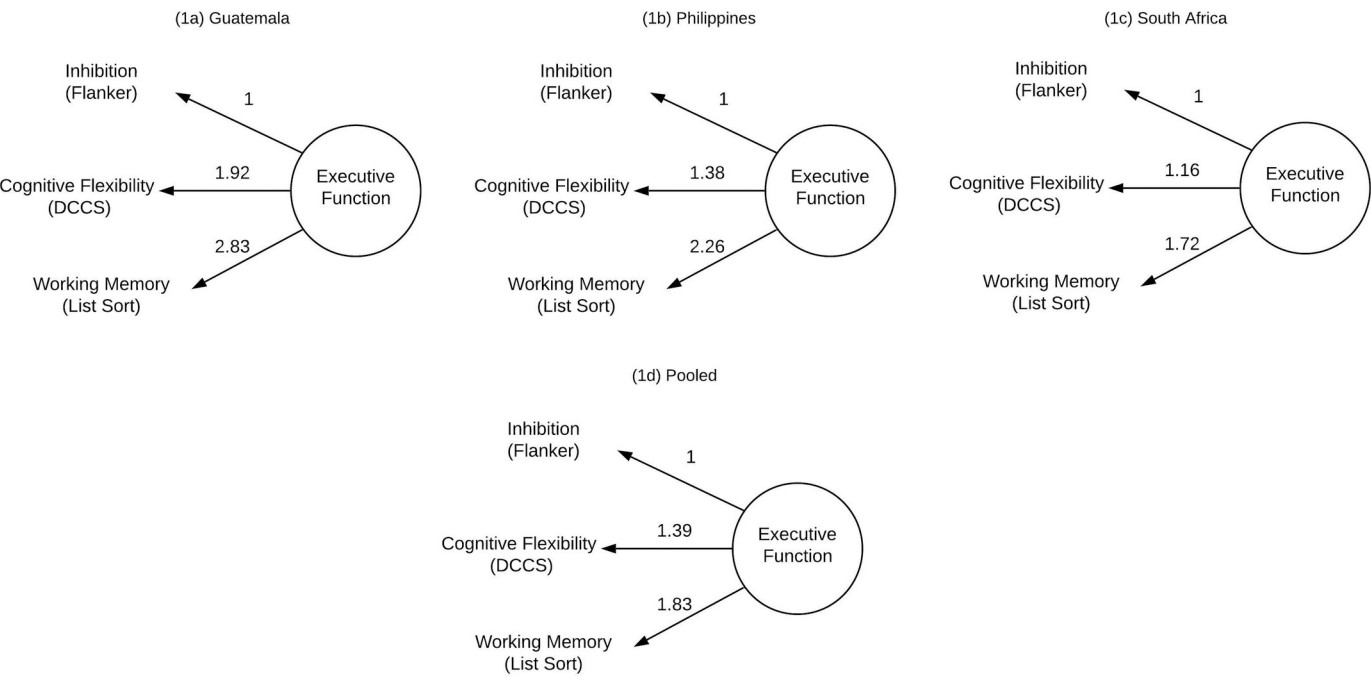

**Fig 1.** Unstandardized Factor loadings of each variable for (a) Guatemala (b) Philippines (c) South Africa and (d) pooled.

**Table 4. Correlation between each country's Executive Function (EF) factor scores and other cognitive measures and schooling.**

| | Guatemala | Philippines | South Africa |
|---|---|---|---|
| | EF Factor Score | EF Factor Score | EF factor score |
| Processing Speed (NIH Pattern comparison) | 0.57* | 0.43* | 0.48* |
| Fluid Intelligence (Raven's Progressive Matrices) | 0.58* | 0.52* | 0.49* |
| Schooling (Highest Grade) | 0.55* | 0.47* | 0.29* |

*p<0.01.

exceptions to this. In the Philippines sample the ICC for the DCCS task was borderline poor-moderate at 0.48, however this lower ICC was mainly driven by one outlier, which when removed changed the ICC to 0.66 (CI = 0.40–0.82), indicating moderate reliability. In addition, within the South African sample the List sorting task was borderline poor-moderate for T1-T2, however again this low ICC was driven by an outlier, which when removed increased the ICC to within the moderate range at 0.59 (CI = 0.35–0.76).

## Discussion

We aimed to explore the feasibility, validity and reliability of EF assessments by exploring factor structure, relationships with other cognitive indices and test-retest reliability of EF measures in three large and diverse low- and middle-income birth cohorts, varying in language, age and culture. Our findings indicate a similar factor structures for each of the three cohorts. In addition, EF factor scores were significantly associated with other measures of cognition as well as schooling; and good test-retest reliability was observed, providing support for the use of these measures cross-culturally.

The factor analysis indicates that, despite cultural and linguistic differences, all three EF measures robustly map onto one factor for all three cohorts. Thus providing support for Miyake's unitary model of EF [1]. Of note, factor loadings were significantly different by site. In large cross-cultural samples such as ours, we would expect some level of statistically significant differences across sites because of language, age, ethnicity and other cultural factors. What is interesting is that, despite these differences, we see similarities in model fit and the measures still hang together as one factor in each site. Our findings serve the very useful purpose of assessing the structure of EF in three large non-WEIRD samples. This is important, given the increasing awareness that cognitive models cannot be tested exclusively in high income countries, and therefore possibly do not generalise to the majority of the world population [13].

**Table 5. Mean (SD) for assessment scores over time for Guatemala.**

| Guatemala | Mean T1 | Mean T2 | t | p | d |
|---|---|---|---|---|---|
| | (n = 45) | (n = 45) | | | |
| Flanker Computed Score | 6.02 (1.27) | 6.53 (1.28) | 5.11 | <0.001 | 0.40 |
| DCCS Computed Score | 5.86 (2.09) | 6.30 (1.88) | 3.08 | <0.01 | 0.22 |
| List Sort Raw Score | 13.50 (4.08) | 14.47 (4.26) | 2.25 | <0.05 | 0.23 |
| Pattern Comparison | 34.64 (10.28) | 36.82 (9.10) | 2.77 | <0.05 | 0.02 |
| Raven's Progressive Matrices | 16.37 (5.99) | 17.49 (6.81) | 2.21 | <0.05 | 0.17 |

**Table 6. Mean (SD) for assessment scores over time for Philippines.**

| Philippines | Mean (SD) T1 | Mean (SD) T2 | t | p | d |
|---|---|---|---|---|---|
| | (n = 32) | (n = 32) | | | |
| Flanker Computed Score | 7.50 (0.99) | 7.66 (0.89) | 1.49 | 0.145 | 0.17 |
| DCCS Computed Score | 7.39 (1.34) | 7.82 (0.79) | 2.32 | <0.05 | 0.39 |
| List Sort Raw Score | 13.42 (4.66) | 14.48 (4.84) | 1.73 | 0.095 | 0.34 |
| Pattern Comparison | 39.25 (9.43) | 41.41 (8.60) | 1.70 | 0.099 | 0.28 |
| Raven's Progressive Matrices | 27.81 (11.79) | 31.19(12.40) | 3.23 | <0.01 | 0.23 |

Given the sparsity of studies exploring EF measurement in LMICs [18], our findings provide crucial evidence that these tasks reliably measure EF across multiple and very different contexts cross-culturally. This EF factor score was also significantly associated with speed of processing, and fluid intelligence, which is consistent with data from HICs [22–25].

Furthermore, a key strength of this study is the significant association between the EF factor score and schooling. This was consistently observed for all three cohorts, supporting the notion that EF and schooling are strongly related [2–6, 14–16] and that these measures translate very well across LMICs. Longitudinal studies of EF and schooling indicate that EF are important components for academic achievement [3, 14–16]. We were not able to establish the causal relationship between EF and schooling through the current study, as EF data was collected for the first time in adulthood. Further longitudinal studies exploring the relationships between childhood EF, schooling, and adult EF would provide interesting insights into the influence of early EF on adult outcomes and may help inform interventions.

The development of executive function can be both positively and negatively affected by the environment within which a person grows up [35]. For individuals in LMICs, where rates of poverty and adverse childhood experiences are significantly higher [11], having robust measures of EF, that relate to key indicators of achievement, can enable researchers to fully capture the pathways to successful adult outcomes and thereby inform policy.

The EF measures showed good reliability in all three countries, demonstrating the robustness of these measures in LMIC settings. Although some small practice effects were observed, this is not unexpected, given the relatively short period of time between tests and retest. These findings are also in line with the NIH validation study, which reports small significant practice effects in US populations [19]. However, this does suggest that studies using NIH Toolbox assessments for multiple assessments should consider adjusting for such effects and /or intervals between sessions.

**Table 7. Mean (SD) for assessment scores over time for South Africa.**

| South Africa | Mean T1 | Mean T2 | Mean T3 | Time | t | p | d |
|---|---|---|---|---|---|---|---|
| | (n = 43) | (n = 43) | (n = 30) | Point | | | |
| Flanker Computed Score | 7.35 (1.16) | 7.61 (0.95) | 7.87 (0.83) | T1-T2 | 2.11 | <0.05 | 0.24 |
| | | | | T2-T3 | 1.98 | 0.058 | 0.29 |
| DCCS Computed Score | 7.62 (0.89) | 7.61 (7.74) | 7.85 (0.63) | T1-T2 | -0.12 | 0.903 | 0.00 |
| | | | | T2-T3 | 2.04 | 0.051 | 0.04 |
| List Sort Raw Score | 13.84 (3.21) | 14.74 (2.67) | 14.93 (3.43) | T1-T2 | 2.03 | <0.05 | 0.30 |
| | | | | T2-T3 | 0.59 | 0.559 | 0.06 |
| Pattern Comparison | 40.91 (6.92) | 42.67 (8.44) | 46.97 (7.38) | T1-T2 | 1.73 | 0.091 | 0.04 |
| | | | | T2-T3 | 6.03 | <0.001 | 0.29 |
| Raven's Progressive Matrices | 33.41 (11.07) | 37.19 (8.99) | 35.24 (12.55) | T1-T2 | 1.58 | 0.123 | 0.37 |
| | | | | T2-T3 | -0.28 | 0.780 | -0.18 |

**Table 8. Intraclass Correlation Coefficients (ICC) with 95% confidence intervals between time points for each measure.**

|  | Flanker Computed Score | DCCS Computed Score | List Sorting | Pattern Comparison | Raven's Progressive Matrices |
|---|---|---|---|---|---|
| Guatemala | 0.76 | 0.80 | 0.71 | 0.90 | 0.86 |
|  | (0.61–0.86) | (0.66–0.88) | (0.52–0.83) | (0.83–0.95) | (0.75–0.92) |
| Philippines | 0.79 | 0.48 | 0.62 | 0.67 | 0.85 |
|  | (0.61–0.89) | (0.17–0.71) | (0.34–0.80) | (0.43–0.82) | (0.71–0.92) |
| South Africa | 0.68 | 0.64 | 0.48 | 0.61 | 0.69 |
| T1-T2 | (0.48–0.81) | (0.43–0.79) | (0.22–0.68) | (0.38–0.77) | (0.47–0.83) |
| South Africa | 0.76 | 0.60 | 0.59 | 0.79 | 0.81 |
| T2-T3 | (0.55–0.88) | (0.32–0.79) | (0.30–0.78) | (0.61–0.89) | (0.64–0.91) |

Note: ICC: <0.50 poor, 0.50–0.75 moderate, 0.75–0.90 good and >0.90 excellent.

The observed practice effects could be due to a number of reasons, such as increased familiarity with the goals of the test, or familiarity with the testing equipment. The latter, in particular, may have been a factor in the current study. Although the cohorts have been participating in research throughout their lives and are familiar with research testing, many of the participants in the Guatemala cohort do not regularly use smartphones or tablets, so completing tasks on an iPad was a novel activity for them. As a result, some participants may have felt more confident in using a tablet to complete the tasks on the second assessment. However, we also report high percentages of participants completing all of the tablet-based EF assessments, suggesting even minimal practice with tablets was not a barrier to completing assessments the first-time round.

Minor adaptations were made to the NIH Toolbox EF tasks to make them culturally appropriate for each of the cohorts. For example, in Guatemala pumpkin stimuli was substituted with papaya, as pumpkins are not native to Guatemala. Substitutions such as these ensure that tasks are culturally appropriate, and participants are not disadvantaged by complete tasks with unfamiliar stimuli. Although we were not able to directly measure the impact of these adaptations on test performance, it is possible that such adaptations contributed to the observation that these tasks measured the same underlying construct in each country. Future research, further exploring the use of the NIH Toolbox EF assessments in LMICs, is needed to determine the extent to which cultural adaptations are necessary in different countries to ensure that the tasks continue to assess the same underlying mechanisms cross-culturally.

Through this study, we also demonstrate that this battery of EF assessments can be administered in LMICs by local field teams. Whilst the local field teams were trained in the administration of cognitive assessments, these teams had not previously received any specialist training. This highlights that, under the supervision of a local psychologist, these tasks can be administered by non-specialist research staff, making them more feasible to employ in low-income settings, overcoming some of the difficulties previously highlighted for other tools that require in-depth training (see Semrud-Clikeman et al., 2017). This supports similar task shifting programs used in mental health research, which demonstrate that the use of non-specialist staff is effective in mental health treatment in LMICS [36] and help reduce the impact of shortages in specialists in LMICs [37, 38].

In summary, when using adapted versions of the NIH Toolbox executive function measures, we observed similar factor structures in all three LMICs, in addition to good test-retest reliability. Furthermore, the EF measures appear to be related to non-verbal IQ, processing speed and also schooling. We demonstrate that these tasks can be successfully administered by local, non-specialist field teams, making them a feasible option for low-income settings. Given

that the three cohorts come from three different continents, with three very different cultural and socio-economic backgrounds, and speak three different languages, these findings have huge implications for the assessment of EF in general and the use of the NIH Toolbox in particular across different countries, cultures and continents.

## Acknowledgments

We would like to thank all of the participants in this study. With thanks to all members of the COHORTS Group. Additional members of the COHORTS group include:

**Pelotas Birth Cohorts**: Fernando C Barros, Fernando P Hartwig, Bernardo L Horta, Ana M B Menezes, Joseph Murray, Fernando C Wehrmeister, Cesar G Victora; **Birth to Twenty Plus**: Shane A Norris, Lukhanyo Nyati; **New Delhi Birth Cohort**: Santosh K Bhargava, Caroline HD Fall, Clive Osmond, Harshpal Singh Sachdev; **INCAP Nutrition Supplementation Trial Longitudinal Study:** Maria F. Kroker-Lobos, Reynaldo Martorell, Manuel Ramirez-Zea; **Cebu Longitudinal Health and Nutrition Survey:** Linda S Adair, Isabelita Bas, Delia Carba, Tita Lorna Perez.

## Author Contributions

**Conceptualization:** Charlotte Wray, Alysse Kowalski, Feziwe Mpondo, Laura Ochaeta, Delia Belleza, Ann DiGirolamo, Rachel Waford, Linda Richter, Nanette Lee, Gaia Scerif, Aryeh D. Stein, Alan Stein.

**Formal analysis:** Charlotte Wray, Alysse Kowalski.

**Funding acquisition:** Linda Richter, Nanette Lee, Aryeh D. Stein, Alan Stein.

**Investigation:** Alan Stein.

**Methodology:** Charlotte Wray, Alysse Kowalski, Feziwe Mpondo, Laura Ochaeta, Delia Belleza, Ann DiGirolamo, Rachel Waford, Linda Richter, Nanette Lee, Gaia Scerif, Aryeh D. Stein, Alan Stein.

**Project administration:** Charlotte Wray, Feziwe Mpondo, Laura Ochaeta.

**Supervision:** Linda Richter, Nanette Lee, Gaia Scerif, Aryeh D. Stein, Alan Stein.

**Validation:** Charlotte Wray.

**Visualization:** Charlotte Wray, Alysse Kowalski.

**Writing – original draft:** Charlotte Wray.

**Writing – review & editing:** Charlotte Wray, Alysse Kowalski, Feziwe Mpondo, Laura Ochaeta, Delia Belleza, Ann DiGirolamo, Rachel Waford, Linda Richter, Nanette Lee, Gaia Scerif, Aryeh D. Stein, Alan Stein.

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
