## [Decision Letter · Decision Letter 0]

4 Aug 2020

PONE-D-20-18734

Executive functions form a single construct and are associated with schooling: evidence from three low- and middle- income countries

PLOS ONE

Dear Dr. Wray,

Thank you for submitting your manuscript to PLOS ONE. After careful consideration, we feel that it has merit but does not fully meet PLOS ONE’s publication criteria as it currently stands. Therefore, we invite you to submit a revised version of the manuscript that addresses the points raised during the review process.

The manuscript has been evaluated by three independent Reviewers, please see their comments appended at the bottom of this letter. As you will see, there were several major concerns that should e addressed in a reviewed version of your study.

We look forward to receiving your revised manuscript.

Kind regards,

Angel Blanch, Ph.D.

Academic Editor

PLOS ONE

Journal Requirements:

3. Please note that in order to use the direct billing option the corresponding author must be affiliated with the chosen institute. Please either amend your manuscript or remove this option (via Edit Submission).

4. We note you have included a table to which you do not refer in the text of your manuscript. Please ensure that you refer to Table 5 in your text; if accepted, production will need this reference to link the reader to the Table.

Reviewers' comments:

Reviewer's Responses to Questions

**Comments to the Author**

1. Is the manuscript technically sound, and do the data support the conclusions?

Reviewer #1: Partly

Reviewer #2: Partly

Reviewer #3: Yes

2. Has the statistical analysis been performed appropriately and rigorously? 

Reviewer #1: No

Reviewer #2: Yes

Reviewer #3: Yes

3. Have the authors made all data underlying the findings in their manuscript fully available?

Reviewer #1: No

Reviewer #2: Yes

Reviewer #3: No

4. Is the manuscript presented in an intelligible fashion and written in standard English?

Reviewer #1: Yes

Reviewer #2: Yes

Reviewer #3: Yes

5. Review Comments to the Author

Reviewer #1: The authors indicate that the instruments to measure Executive Functions have been validated mainly in WEIRD samples. The goal of the paper is to address this limitation by employing an impressive database of individuals from Guatemala, the Philippines, and South Africa. The article is in general very well written. The main problem I see is that the statistical methods used are not completely suitable. Below are some suggestions for improvement that I hope the authors will find useful.

1. The statistical procedure that allows the evaluation of the generalizability of the instruments across cultures, language or other variables is the measurement invariance analysis. The works of Bach et al. (2018) and Marsh et al. (2009) illustrate how this can be done using an exploratory factor analysis approach. The separate analysis in each database does not allow us to say that the measured construct does not vary. The analysis provided by the authors could be understood in any case as a modest first contribution to the study of the measurement invariance of this test.

Work cited:

Bach, B., Sellbom, M., & Simonsen, E. (2018). Personality inventory for DSM-5 (PID-5) in clinical versus nonclinical individuals: Generalizability of psychometric features. Assessment, 25(7), 815-825.

Marsh, H. W., Muthén, B., Asparouhov, T., Lüdtke, O., Robitzsch, A., Morin, A. J., & Trautwein, U. (2009). Exploratory structural equation modeling, integrating CFA and EFA: Application to students' evaluations of university teaching. Structural equation modeling: A multidisciplinary journal, 16(3), 439-476.

2. With respect to the exploratory factor analysis included, it would be advisable to include all 9 items in the same analysis. The degrees of freedom would be positive and the fit of the model could be assessed. In addition, the correlations between the scores obtained with oblique rotation could be reported. It is also necessary to indicate which estimation method was used (e.g., ML, ULS). Finally, the use of the scree plot has been criticized because its interpretation is highly subjective. Authors should use the parallel analysis method, as it is the gold standard method for dimensionality assessment (Auerswald & Moshagen, 2019; Finch, 2020; Lim & Jahng, 2019).

Work cited:

Auerswald, M., & Moshagen, M. (2019). How to determine the number of factors to retain in exploratory factor analysis: A comparison of extraction methods under realistic conditions. Psychological Methods, 24, 468–491.

Finch, W. H. (2020). Using fit statistics differences to determine the optimal number of factors to retain in an exploratory factor analysis. Educational and Psychological Measurement, 80, 217–241.

Lim, S., & Jahng, S. (2019). Determining the number of factors using parallel analysis and its recent variants. Psychological Methods, 24, 452–467.

3. In Table 3 the authors calculate the correlation of the total test score (i.e. considering the three factors) and several criterion variables. However, this factorial model has not been tested. For it to make sense to use the total factor scores it would be necessary to show 1) that a unidimensional for all items obtains good fit, 2) that a three-factor model with all items obtains good fit and the factors are highly correlated. Without doing so, there is some inconsistency between this analysis and the factor analysis presented before.

4. I appreciate that the authors report the test-retest reliability statistic, but it would be necessary to also include Cronbach's alpha and, the most accepted, Omega coefficient (Viladrich, Angulo-Brunet, & Doval, 2017).

Work cited:

Viladrich, C., Angulo-Brunet, A., & Doval, E. (2017). A journey around alpha and omega to estimate internal consistency reliability. Annals of Psychology, 33(3), 755-782.

5. An effort should be made to clarify the goal of the study. For example, the title seems to indicate that the relationship with schooling will be paramount, in the introduction it is important to determine whether the factor structure in non-WEIRD samples is similar to that obtained with WEIRD samples, and in the results section much of the text is devoted to commenting on the results for practice effects. Moreover, it is necessary to discuss the results in relation to those obtained with WEIRD samples. What previous results on factor structure can be commented on? What has been the correlation obtained with schooling?

Reviewer #2: The paper focuses on a new area of investigation, the executive functions and schooling in three low- and middle- income countries. The topic is interesting, but the analyses are poor, the design of the study is not clear and the results appear limited.

It is not clear the real objectives of this study and the valued added of it.

I could give some suggestions to ameliorate it.

1. It is not clear the number of control group participants. How did you decide this number? Have you run power analysis? Not clear why you showed these control groups and what is their function in your study.

2. You didn’t show in a table the participants’ characteristics

3. You also noted that It could be present a memory effect in the test-retest of the FE tasks with a so less number of days between the two assessments. Why did you decide to have this measure? Clarify it.

4. What about possible gender or age differences? It would be interesting to have also this information to have a complete assessment of this EF measure in these LMI countries

5. I don’t found the tables with participants’ characteristics in each country. You stressed the schooling, but you didn’t report it in the tables and you didn’t stress the schooling situation in these countries and why it is important for the aim of the study.

6. The discussion could be more interesting adding the new suggested analysis and with the revisions suggested below.

Reviewer #3: Review of “Executive functions form a single construct and are associated with schooling: Evidence from three low- and middle- income countries” by Wray et al. for PLOS ONE PONE-D-20-18734

The present research reports a study conducted in Guatemala, the Philippines and South Africa measure the executive function (EF) in three large adults samples. The goal of the research is to demonstrate that the three target measures (a flanker task, the DCCS, and a list sort task) cohere well in assessing EF. The work does demonstrate this; it also illustrates that EF assessment in low and middle-income countries can be achieved without highly trained personnel, but with the help of portable technology (tablets). The data show good convergent validity, good to reasonable test-retest reliability, but also modest training effects.

Overall, this paper does provide good evidence for what it sets out to demonstrate, and will be solid evidence that EF assessment is feasible in low to middle-income countries. Whereas some researchers may have taken this for granted all along, the present research certainly will assuage the concerns of those who have doubted this. In this sense, this paper will certainly be useful to the field.

There are few observations that would be helpful to have the authors address, though.

1. The Guatemala sample is older and more heterogeneous than the other two samples in terms of age, and the South African samples appears to be most homogeneous (no age variation). In Guatemala the age of participants ranges from 38 to 58 years, a time during which much can happen in the development of a society. This might be in part responsible why the correlation coefficient for Guatemala is highest for schooling and lowest for South Africa. Do findings hold when the Guatemalan samples is divided in an older and a younger half? This also gets at possible age effects that one might expect within the Guatemala sample.

2. Ideally, when the factorial structure is to be compared across different samples, one would expect the authors to conduct a confirmatory factor analysis, demonstrating the convergence across the three country samples. In the present scenario, one would expect that the model would not fit very well, simply because with large samples even small differences in factor loadings would undermine the model fit (such as the fact that the loading of working memory task is much higher in the Guatemalan sample than the other two samples). Doing this may not be too critical. However, the reader might want to see one additional decimal for the factor loadings in Figure 1 (for a total of two decimals), as the fit might be reasonably good. Yet, coefficients might not be identical for the flanker task and DCCS, as currently implied.

3. Small issue: In Table 5, the performance decreases between T1 and T2 for the DCCS, and between T2 and T3 for the Raven’s progressive matrices. The corresponding t- and d-values should be negative.

6. PLOS authors have the option to publish the peer review history of their article (what does this mean?). If published, this will include your full peer review and any attached files.

Reviewer #1: No

Reviewer #2: **Yes: **Marta Tremolada

Reviewer #3: **Yes: **Markus Kemmelmeier

---

## [Author Response · Author response to Decision Letter 0]

28 Oct 2020

Editor Comments

Comment 1. Please ensure that your manuscript meets PLOS ONE's style requirements, including those for file naming. The PLOS ONE style templates can be found at

Response: Done

Comment 2. We note that you have indicated that data from this study are available upon request. PLOS only allows data to be available upon request if there are legal or ethical restrictions on sharing data publicly. For more information on unacceptable data access restrictions, please see http://journals.plos.org/plosone/s/data-availability#loc-unacceptable-data-access-restrictions.

Response: 

Please see below our updated data availability statement:

We are unable to make the data publicly available because the data come from active cohort studies with previously published recruitment information and hence there is potential for individual participant re-identification if the raw data are made freely available. Data are available to qualified researchers upon request, subject to confidentiality agreements. Data requests can be made to the following:

South Africa data: Linda Richter, Linda.Richter@wits.ac.za or Shane Norris Shane.Norris@wits.ac.za

Guatemala data: Manuel Ramierez, mramirez@incap.int or Dina Roche droche@incap.int

Philippines data: Nanette Lee, nanette_rlee@yahoo.com or Delia Carba carbadel@yahoo.com

Additional data comments:

In response to the follow up comments about our data availability statement, please see below answers:

Additional comment 1. Who is restricting the data (e.g. IRB, ethics committee, etc.)?

Response: The data are restricted based on limited data use agreements established by the data owners.

Additional comment 2. Can an anonymous dataset be made available?

Response: The data cannot be fully anonymized as the cohorts themselves are well described and not large enough to effectively anonymize to prevent re-identification.

Additional comment 3. Please provide an institutional, non-author point of contact where we can direct data inquiries. Note that it is not acceptable for the authors to be the sole named individuals responsible for ensuring data access. If data are owned by a third party, please indicate how others may request data access.

Response: Data inquiries can be made to :

South Africa data: Linda Richter, Linda.Richter@wits.ac.za or Shane Norris Shane.Norris@wits.ac.za

Guatemala data: Manuel Ramierez, mramirez@incap.int or Dina Roche droche@incap.int

Philippines data: Nanette Lee, opsfoundation@opsusc.org or Delia Carba opsfoundation@opsusc.org

Additional comment 4. Thank you for providing alternate points of contact for your data. We notice you have provided personal email addresses for some of the data. In the interest of making data available to future researchers, please provide details on how you will ensure persistent or long-term data storage and availability. For instance, data might be stored in two independent locations. We will then update your Data Availability Statement.

Response: We have updated the email address with a central institutional email (see above). Each site is responsible for the long-term maintenance and back-up of their data. 

Comment 3. Please note that in order to use the direct billing option the corresponding author must be affiliated with the chosen institute. Please either amend your manuscript or remove this option (via Edit Submission).

 Response: We are happy for the direct billing option to be removed. We have contacted you regarding this, as we are unable to see the billing options in the resubmission form, and were advised to include it in the cover letter so that this could be changed one we had submitted our revised manuscript. 

Comment 4. We note you have included a table to which you do not refer in the text of your manuscript. Please ensure that you refer to Table 5 in your text; if accepted, production will need this reference to link the reader to the Table.

Response: We have now added this to the text on pg 15.

Review Comments to the Author

Reviewer #1: 

Comment 1. The statistical procedure that allows the evaluation of the generalizability of the instruments across cultures, language or other variables is the measurement invariance analysis. The works of Bach et al. (2018) and Marsh et al. (2009) illustrate how this can be done using an exploratory factor analysis approach. The separate analysis in each database does not allow us to say that the measured construct does not vary. The analysis provided by the authors could be understood in any case as a modest first contribution to the study of the measurement invariance of this test.

Response: Many thanks for this suggestion, which we have taken on board; we have now included measurement invariance analysis in the manuscript. However, we note that our primary aim was not to establish complete equivalence across cohorts: we explored the extent to which these measures could be employed in these differing settings, with limited attrition, and how individual differences across observed measures related to each other across these large and diverse samples. Weak invariance by site could not be established (see Table 3). However, given the heterogeneity in ages, languages, and contexts across countries, we do not think this is wholly unexpected. We also note that, as Reviewer #3 points out, “with large samples even small differences in factor loadings would undermine the model fit (such as the fact that the loading of working memory task is much higher in the Guatemalan sample than the other two samples)”, suggesting that our very large sample sizes were likely to point towards variance by site. We have added a discussion of this point in the general discussion. 

Comment 2. With respect to the exploratory factor analysis included, it would be advisable to include all 9 items in the same analysis. The degrees of freedom would be positive and the fit of the model could be assessed. In addition, the correlations between the scores obtained with oblique rotation could be reported. It is also necessary to indicate which estimation method was used (e.g., ML, ULS). Finally, the use of the scree plot has been criticized because its interpretation is highly subjective. Authors should use the parallel analysis method, as it is the gold standard method for dimensionality assessment (Auerswald & Moshagen, 2019; Finch, 2020; Lim & Jahng, 2019).

Response: In response to both reviewer 1 and reviewer 2 comments regarding the factor analysis we have made the following revisions:

To explore the structure of the three NIH toolbox executive function measures (cognitive inhibition, cognitive flexibility, and working memory) we conducted exploratory factor analysis (EFA) and confirmatory factor analysis (CFA) in pooled and site-specific samples. 

To inform the number of factors specified in the EFA we performed a parallel analysis in pooled and site-specific samples which indicated that a one factor solution best fit the data, and all one factor models were robust. 

We have also now included a section indicating how each of the three measures, tapping diverse executive functions, correlate with each other. 

In an effort to maximize data, we have switched from using maximum likelihood to full information maximum likelihood estimation. As a result, our sample sizes are now slightly larger, and associations of the EF factor score with other variables have changed slightly (see table 4).

Comment 3. In Table 3 the authors calculate the correlation of the total test score (i.e. considering the three factors) and several criterion variables. However, this factorial model has not been tested. For it to make sense to use the total factor scores it would be necessary to show 1) that a unidimensional for all items obtains good fit, 2) that a three-factor model with all items obtains good fit and the factors are highly correlated. Without doing so, there is some inconsistency between this analysis and the factor analysis presented before.

Response: We think generally there has been some confusion about the structure of our model. We administered three assessments that are indicators of the executive function latent construct. Each assessment measures a unique aspect of executive function. These include the Flanker task, which measures inhibition; the Dimensional Change Card Sort, which measures flexibility; and the List Sort, which measures working memory. We use the overall score from each of these assessments as indicators of executive function.

We recognise that we could have explained our model more thoroughly and we have now included further information in the methods regarding our analysis, along with an explanation of the additional analyses we have now included as a result of the reviewer comments.

We have now included correlations in the manuscript between each of the EF instruments’ total scores. 

Comment 4. I appreciate that the authors report the test-retest reliability statistic, but it would be necessary to also include Cronbach's alpha and, the most accepted, Omega coefficient (Viladrich, Angulo-Brunet, & Doval, 2017).

Response: The Omega coefficient uses item-level data which is not applicable to our data. Our executive function assessments ask participants to perform a task over multiple trials. They are scored accounting for accuracy and reaction time and the resulting computed score is what we have used in the analysis.

Comment 5. An effort should be made to clarify the goal of the study. For example, the title seems to indicate that the relationship with schooling will be paramount, in the introduction it is important to determine whether the factor structure in non-WEIRD samples is similar to that obtained with WEIRD samples, and in the results section much of the text is devoted to commenting on the results for practice effects. Moreover, it is necessary to discuss the results in relation to those obtained with WEIRD samples. What previous results on factor structure can be commented on? What has been the correlation obtained with schooling?

Response: We recognise that we could have made the aims of the paper clearer and also included more information regarding the importance of education in this paper. We have now included more specific information in the introduction and discussion regarding both of these points. 

Reviewer #2: 

Comment 1. It is not clear the number of control group participants. How did you decide this number? Have you run power analysis? Not clear why you showed these control groups and what is their function in your study.

Response: We are unsure what reviewer 2 means by control group participants, as we do not have a control group in this study. In this study the factor analyses are run on the whole cohort (n’s for this group are presented in the cohort description and again in table 1). The test-retest reliability was conducted on a smaller group of participants, N’s are presented in the methods section in addition to tables 5, 6 and 7.

Comment 2. You didn’t show in a table the participants’ characteristics.

Response: Participant are described in the cohort description (pg6). In addition information about NV-IQ and schooling are presented in table 1, along with means for all of the measures. 

Comment 3. You also noted that It could be present a memory effect in the test-retest of the FE tasks with a so less number of days between the two assessments. Why did you decide to have this measure? Clarify it.

Response: We tested how robust the scores were using test-retest reliability and practice effects as this is a commonly used way to do this. This is also the method used in the NIH validation study (19). In the discussion we state that we expect practice effects given the relatively short period of time between tests and retest. We also highlight that these findings are also in line with the NIH validation study, which reports small significant practice effects in US populations (19). We also discuss the potential limitations of using these kinds of tasks for repeated administration, given the small practice effects observed. 

Comment 4. What about possible gender or age differences? It would be interesting to have also this information to have a complete assessment of this EF measure in these LMI countries.

Response: Our cohorts all vary in age. As such, the observation of similar factor structure and reliability of these EF measures across the cohorts does provide some information about age. Factor loadings do not differ by sex (not reported in the manuscript, for brevity).

Comment 5. I don’t found the tables with participants’ characteristics in each country. You stressed the schooling, but you didn’t report it in the tables and you didn’t stress the schooling situation in these countries and why it is important for the aim of the study.

Response: Participant characteristics such as age are described in the cohort descriptions on pg. 6. Information about NV-IQ and schooling are presented in table 1, along with means for all of the measures. We have now revised the introduction to strengthen the aims of the study. 

Comment 6. The discussion could be more interesting adding the new suggested analysis and with the revisions suggested below.

Response: We have made changes to both the analysis and discussion sections in response to all of the reviewer comments. Please see the additions made to these sections. 

Reviewer #3: 

Comment 1. The Guatemala sample is older and more heterogeneous than the other two samples in terms of age, and the South African samples appears to be most homogeneous (no age variation). In Guatemala the age of participants ranges from 38 to 58 years, a time during which much can happen in the development of a society. This might be in part responsible why the correlation coefficient for Guatemala is highest for schooling and lowest for South Africa. Do findings hold when the Guatemalan samples is divided in an older and a younger half? This also gets at possible age effects that one might expect within the Guatemala sample.

Response: Age is an interesting variable to consider in this study. Each of our cohorts represent participants of different ages. Within each site there are no meaningful age differences. There is more variation in Guatemala, but the mean difference is ~7 years, which is small relative to the differences in age across sites and it is unlikely this difference will be significant. 

Some of the differences in factor loadings across sites could be a driven by the different ages of the cohorts. However, as there are many aspects (e.g. language, culture, SES) that could impact site differences, we cannot specify the unique contribution of age in this study.

Comment 2. Ideally, when the factorial structure is to be compared across different samples, one would expect the authors to conduct a confirmatory factor analysis, demonstrating the convergence across the three country samples. In the present scenario, one would expect that the model would not fit very well, simply because with large samples even small differences in factor loadings would undermine the model fit (such as the fact that the loading of working memory task is much higher in the Guatemalan sample than the other two samples). Doing this may not be too critical. However, the reader might want to see one additional decimal for the factor loadings in Figure 1 (for a total of two decimals), as the fit might be reasonably good. Yet, coefficients might not be identical for the flanker task and DCCS, as currently implied.

Response: We have now updated the analysis in response to reviewer 3 and reviewer 1’s comments (see comments above). To explore the factor structure of the NIH toolbox executive function measures we conducted exploratory and confirmatory factor analyses in overall sample (pooled) and by site. 

We have also updated the Figure to include two decimal places and also to indicate factor loadings for the pooled data. Also, to note, we have changed the figures to represent unstandardized factor loadings, as opposed to standardized. 

Comment 3. Small issue: In Table 5, the performance decreases between T1 and T2 for the DCCS, and between T2 and T3 for the Raven’s progressive matrices. The corresponding t- and d-values should be negative.

Response: Table has now been corrected. Due to the addition of tables in the document, this is now table 7.

---

## [Decision Letter · Decision Letter 1]

12 Nov 2020

Executive functions form a single construct and are associated with schooling: evidence from three low- and middle- income countries

PONE-D-20-18734R1

Dear Dr. Wray,

We’re pleased to inform you that your manuscript has been judged scientifically suitable for publication and will be formally accepted for publication once it meets all outstanding technical requirements.

Kind regards,

Angel Blanch, Ph.D.

Academic Editor

PLOS ONE

Additional Editor Comments (optional):

Reviewers' comments:

Reviewer's Responses to Questions

**Comments to the Author**

1. If the authors have adequately addressed your comments raised in a previous round of review and you feel that this manuscript is now acceptable for publication, you may indicate that here to bypass the “Comments to the Author” section, enter your conflict of interest statement in the “Confidential to Editor” section, and submit your "Accept" recommendation.

Reviewer #2: All comments have been addressed

2. Is the manuscript technically sound, and do the data support the conclusions?

Reviewer #2: Yes

3. Has the statistical analysis been performed appropriately and rigorously? 

Reviewer #2: Yes

4. Have the authors made all data underlying the findings in their manuscript fully available?

Reviewer #2: Yes

5. Is the manuscript presented in an intelligible fashion and written in standard English?

Reviewer #2: Yes

6. Review Comments to the Author

Reviewer #2: I found the paper really ameliorated following the reviewer's suggestions. I appreciated your hard work.

7. PLOS authors have the option to publish the peer review history of their article (what does this mean?). If published, this will include your full peer review and any attached files.

Reviewer #2: **Yes: **Marta Tremolada

---

## [Editor Report · Acceptance letter]

16 Nov 2020

PONE-D-20-18734R1 

Executive functions form a single construct and are associated with schooling: evidence from three low- and middle- income countries 

Dear Dr. Wray:

I'm pleased to inform you that your manuscript has been deemed suitable for publication in PLOS ONE. Congratulations! Your manuscript is now with our production department. 

Kind regards, 

on behalf of

Dr. Angel Blanch 

Academic Editor

PLOS ONE